# K-Shaped Silicon Waveguides for Logic Operations at 1.55 μm

**Amer Kotb** [1,2,*] and **Kyriakos E. Zoiros** [3]

1   GPL, State Key Laboratory of Applied Optics, Changchun Institute of Optics, Fine Mechanics, and Physics, Chinese Academy of Sciences, Changchun 130033, China
2   Department of Physics, Faculty of Science, University of Fayoum, Fayoum 63514, Egypt
3   Lightwave Communications Research Group, Department of Electrical and Computer Engineering, School of Engineering, Democritus University of Thrace, 67100 Xanthi, Greece
*   Correspondence: amer@ciomp.ac.cn

**Abstract:** Silicon has properties that make it the preferable semiconductor material for realizing a wide suite of electronic devices. In this paper, all basic optical logic operations, including XOR, AND, OR, NOT, NOR, XNOR, and NAND, are demonstrated by means of simulation using K-shaped compact silicon waveguides operated at the 1.55 μm telecommunications wavelength. This waveguide comprises three waveguide strips, all made of silicon printed on silica. By adjusting the phase of the incident beams, the pursued logic operations can be realized. To evaluate how well the considered operations are performed, the contrast ratio (CR) is employed as a figure of merit. Compared to other reported waveguides, the suggested K-shaped waveguide achieves higher CRs and a speed of the order of 120 Gb/s.

**Keywords:** logic operations; silicon-on-silica waveguide; contrast ratio

## 1. Introduction

By enabling the execution of signal-processing functionalities without troublesome optoelectronic conversions at the photonic nodes, all-optical gates serve as essential building blocks for the construction of lightwave broadband communications networks [1]. The accomplishment of the many signal processing tasks entirely in the optical domain, such as packet processing [2,3], pseudorandom binary sequence generation [1,4], encryption/decryption [5], error detection and correction [6], arithmetic operations [7,8], construction of optical memory elements [9], digital comparison [10,11], buffering [12], implementation of any other Boolean function [13], and construction of combinational logic circuits [14], is made possible by the XOR, AND, OR, NOT, NOR, NAND, and XNOR logic operations. On the other hand, the development of effective and low-loss platforms at a reasonable cost is claimed by silicon photonics. A type of structure known as silicon-on-silica technology is created by depositing a thin layer of crystalline silicon on an insulating layer, which is silica (silicon dioxide). Due to the significant infrared transparency of silicon and refractive index difference between silicon (i.e., core with $n_{silicon} \approx 3.48$ at 1.55 μm) and silica (i.e., cladding with $n_{silica} \approx 1.444$ at 1.55 μm), silicon-on-silica optical waveguides have unique optical features [15]. Various optical waveguides have been recently used for implementing both all-optical logic gates and all-optical networks [16–23]. Therefore, in this paper, we simulate seven basic logic operations, including XOR, AND, OR, NOT, NOR, NAND, and XNOR, using K-shaped waveguides operated at the telecommunications wavelength of 1.55 μm. This waveguide has four terminals, each of which has an output port and three input ports composed of silicon patterned on silica. It is generally known that silicon has a relatively low optical loss (2 dB/cm) for wavelengths up to 8 μm, but silica's optical loss increases rapidly beyond 3.6 μm [15]. The interferences, both constructive and destructive, which are created by the phase difference between the input beams, are the key for the realization of the considered logic operations. In order to demonstrate how the logic operations are

executed, finite-difference time-domain (FDTD) solutions are obtained, using commercially available software, with the convolutional optimally matched layer as an absorbing boundary condition. The logic operations' performance is assessed against the contrast ratio (CR) metric. According to the derived simulation results, the employed waveguide can achieve higher CRs at an extended data rate of 120 Gb/s and, hence, can outperform previously reported designs [16–20].

## 2. K-Shaped Waveguide

In this work, we build a K-shaped waveguide with four terminals made of three silicon slots patterned on a silica substrate. The three input ports are excited by an electromagnetic wave that is polarized in the transverse magnetic mode at 1.55 μm. The wavelength and intensity of the incident beams are identical. The K-shaped silicon-on-silica waveguide is depicted schematically in Figure 1, along with its field intensity distributions.

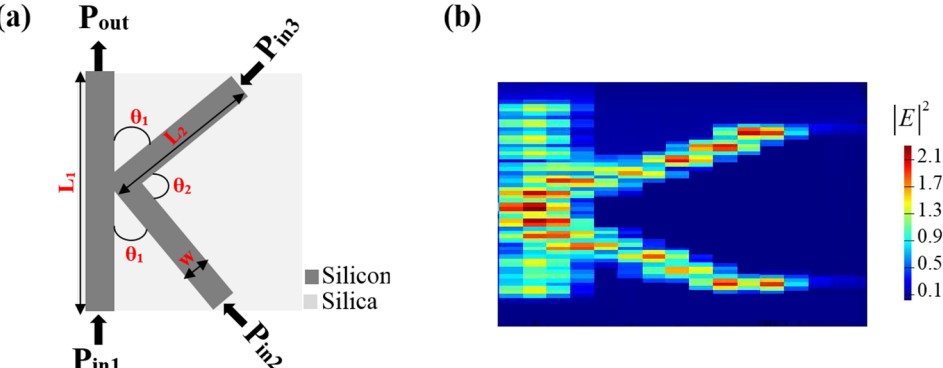

**Figure 1.** (**a**) Schematic depiction and (**b**) field-intensity distributions of K-shaped silicon-on-silica waveguide.

To record the simulation outcomes, the FDTD monitors are employed. Setting the threshold transmission ($T_{th}$) value to 0.12 is necessary at first. The formula for the output transmission (T) is $T = I_{out}/I_{in}$ [16], where $I_{out} = |E_{out}|^2$ is the intensity at $P_{out}$, and $I_{in} = I_1 + I_2 + I_3$ is the sum of the intensities at the three input ports. The input beams must satisfy the requirements for phase-matching in order to maximize T. In essence, this implies ensuring sure that the interacting waves are kept in the proper relative phase throughout the direction of propagation. However, before high CR logic gates can be accomplished, the phase-matching condition necessitates a specific selection of the input wavelength and waveguide characteristics. The phase-matching in silicon waveguides is induced by the contributions of the waveguide birefringence, material dispersion, waveguide dispersion, and cross- and phase-self modulations [24]. It is, therefore, feasible to achieve phase-matching by designing the waveguide such that the birefringence and material dispersion terms cancel one another, according to the phase-matching analysis of silicon waveguides, as reported in [24]. When $T > T_{th}$, $P_{out}$ generates a logical output of '1', while in all other cases, it generates a '0'. The CR is an important metric for logic devices and is defined as $CR(dB) = 10\ ln\left[P^1_{mean}/P^0_{mean}\right]$ [25], where $P^1_{mean}$ and $P^0_{mean}$ represent the mean peak powers of output logic bits '1' and '0', respectively. Compared to other metrics, such as the extinction ratio, the CR offers a better and more accurate evaluation of the performance of the optical logic operations [26]. For the proposed waveguide, Table 1 lists the default parameters' values used in the simulation.

**Table 1.** Simulation parameters.

| Symbol | Definition | Value | Unit |
|:---:|:---:|:---:|:---:|
| $L_1$ | Length of long slot | 2.5 | µm |
| $L_2$ | Length of short slot | 1.0 | µm |
| W | Width of slot | 0.22 | µm |
| D | Thickness of slot | 0.3 | µm |
| $\theta_1$ | Angle between long and short slots | 50 | degree |
| $\theta_2$ | Angle between short slots | 80 | degree |
| $n_{silicon}$ | Silicon refractive index at 1.55 µm | 3.48 | - |
| $n_{silica}$ | Silica refractive index at 1.55 µm | 1.444 | - |
| $\lambda$ | Operating wavelength | 1.55 | µm |
| $T_{th}$ | Threshold transmission | 0.12 | - |

When all incident beams (i.e., two beams and a reference or clock light) are launched at the three input ports with the same phase of 180°, the normalized spectral transmission (T) and loss as a function of the operating wavelength (λ) are shown in Figure 2. The employed waveguide achieves a high T = 0.852 and a low loss = 0.69 dB/µm at 1.55 µm. Such small propagation losses are a direct result of the scattering at the inner slots' interfaces and absorption within the materials. Figure 2 also shows that this waveguide achieves a high T and a low loss at a wide range of telecommunication wavelengths, from 1.3–1.6 µm.

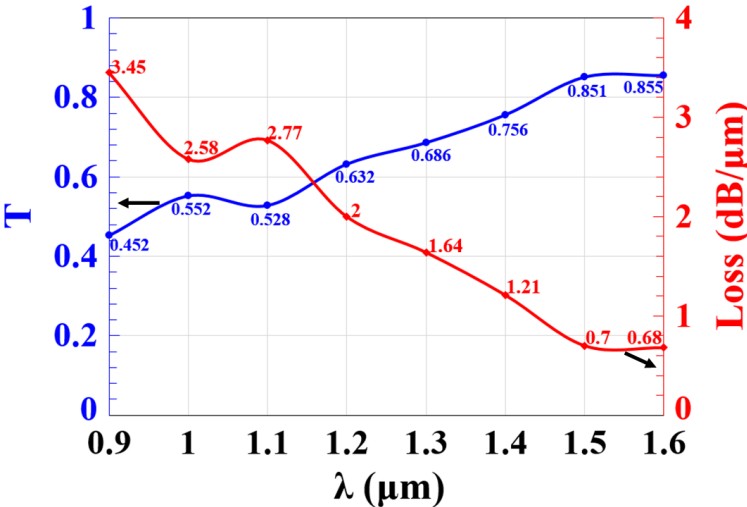

**Figure 2.** Normalized spectral transmission (T) and loss versus operating wavelength (λ), using K-shaped silicon-on-silica waveguide.

Relaxed tolerances are crucial for both manufacturing and operating conditions. Manufacturing tolerances refer to the management of the geometrical dimensions during processing and their ensuing effect on device performance. Operation tolerances describe how the device responds to variations in wavelength, polarization, temperature, input field distribution, and refractive index [27,28]. Most laser sources have a significant practical wavelength tolerance. For example, a 1550 nm fiber laser may have a wavelength tolerance of ±20 nm, resulting in an actual wavelength of 1550 ± 20 nm [29]. Figure 3 shows the dependence of the loss on the wavelength tolerance using the proposed waveguide. This result has been taken based on Equations (4)–(7) from Ref. [30].

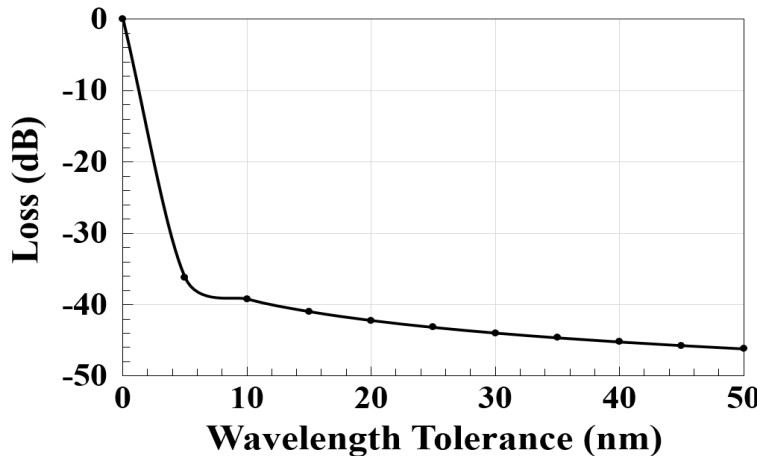

**Figure 3.** Optical loss versus wavelength tolerance, using K-shaped silicon-on-silica waveguide.

In this waveguide, we have used three acute angles, with a sum that is 180°, to perform the K letter. These angles (i.e., $\theta_1$ and $\theta_2$) play an important role in the K-shaped design in order to implement the considered logic gates with high CRs. Thus, the effect of the angle between the long and short slots ($\theta_1$) on a normalized spectral transmission (T) at an operating wavelength of 1550 nm is simulated, as shown in Figure 4. It is clear from Figure 4 that the highest T occurred at $\theta_1 = 50°$ (i.e., $\theta_2 = 80°$), as optimized in this simulation.

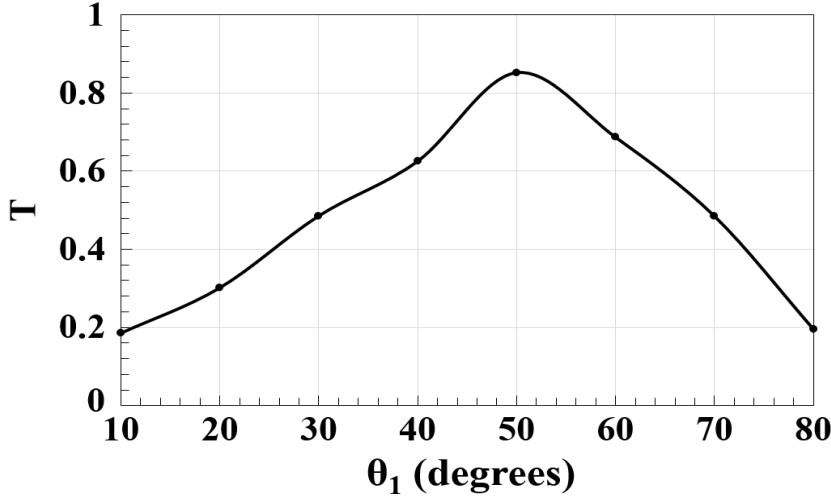

**Figure 4.** Normalized spectral transmission (T) versus angle between long and short slots ($\theta_1$), using K-shaped silicon-on-silica waveguide.

## 3. Logic Operations

### 3.1. XOR

To carry out the XOR, AND, and OR logic operations, a reference beam (REF) must be injected into $P_{in2}$ of Figure 1, while the other two beams are launched into $P_{in1}$ and $P_{in3}$. The REF (all '1's) is used to introduce a reference phase difference between the input signals, resulting in either constructive or destructive interference. Constructive interference happens when all input beams are injected at the same phase (resulting in an output of '1'); destructive interference happens when they are launched at different phases (resulting in an output of '0'). As a result, for an XOR operation, $P_{out}$ produces a '1' (meaning $T > T_{th}$) because of the constructive interference that occurs between the input beams when the combination of these input beams (01, 10) is injected along with the REF at the same phase (i.e., $\Phi_1 = \Phi_3 = \Phi_{REF} = 180°$). The destructive interference between the incident beams causes a '0' output to be produced at $P_{out}$ (meaning $T < T_{th}$) when the

combination (11), with the REF at different phases (i.e., $\Phi_1 = 0°$, $\Phi_3 = 90°$, and $\Phi_{REF} = 180°$), is launched. This results in the XOR logic function. We notice the presence of light at ports having '0' input, which is a natural result because the inner interfaces of the three input ports of the K-shaped waveguide are all opposite, and, therefore, the light is deflected inside them in an outward direction. The XOR field intensity distributions are displayed in Figure 5, using a K-shaped silicon-on-silica waveguide at 1.55 µm.

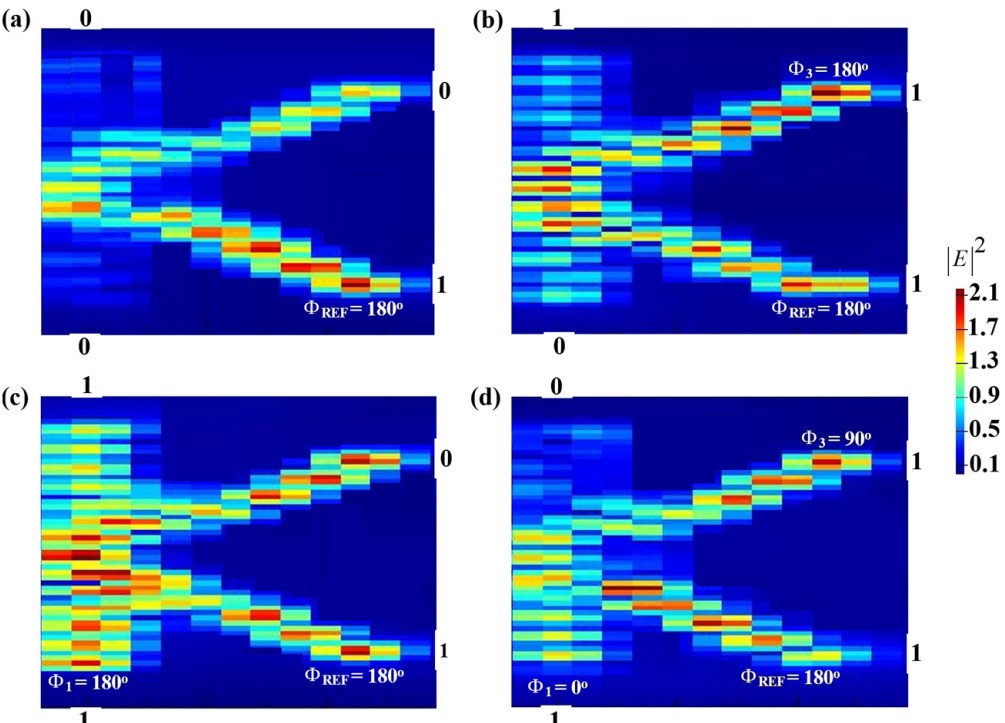

**Figure 5.** XOR field-intensity distributions, using K-shaped silicon-on-silica waveguide at 1.55 µm: (**a**) '00' input, (**b**) '01' input, (**c**) '10' input, and (**d**) '11' input.

Due to the large difference between the mean peak powers of '1' and '0', the suggested waveguide achieves a high CR = 34 dB. The XOR simulation outcomes, employing the K-shaped silicon-on-silica waveguide at 1.55 µm, are shown in Table 2.

**Table 2.** XOR simulation outcomes ($T_{th} = 0.12$).

| $P_{in1}$ | $P_{in3}$ | $P_{in2}$ (REF) | T | $P_{out}$ | CR (dB) |
|-----------|-----------|-----------------|-------|-----------|---------|
| 0 | 0 | 1 | 0.021 | 0 | |
| 0 | 1 | 1 | 0.464 | 1 | |
| 1 | 0 | 1 | 0.852 | 1 | 34 |
| 1 | 1 | 1 | 0.023 | 0 | |

### 3.2. AND

Similar to the XOR operation, the AND operation involves injecting two beams into $P_{in1}$ and $P_{in3}$ as well as launching the REF (all '1's) from $P_{in2}$. $P_{out}$ creates a '1' output, due to constructive interference, when all incident beams are released into the proposed waveguide at the same phase (i.e., $\Phi_1 = \Phi_3 = \Phi_{REF} = 180°$). In contrast, when these incident beams are injected at a different phase, $P_{out}$ outputs a '0' because of destructive interference. This results in the AND operation. In Figure 6, the AND field intensity distributions are shown, using a K-shaped silicon-on-silica waveguide at 1.55 µm.

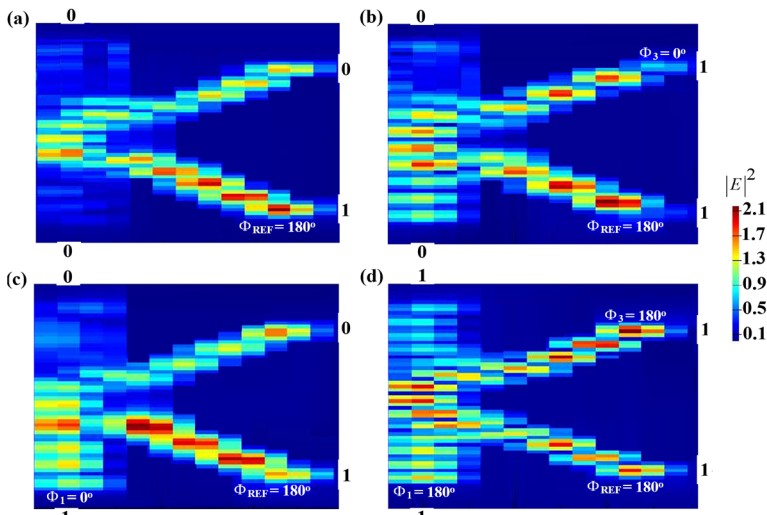

**Figure 6.** AND field intensity distributions, using K-shaped silicon-on-silica waveguide at 1.55 μm: (**a**) '00' input, (**b**) '01' input, (**c**) '10' input, and (**d**) '11' input.

For the logic AND operation, the proposed waveguide achieves CR = 31 dB at 1.55 μm. The further results of the AND simulation are listed in Table 3.

**Table 3.** AND simulation outcomes ($T_{th}$ = 0.12).

| $P_{in1}$ | $P_{in3}$ | $P_{in2}$ (REF) | T | $P_{out}$ | CR (dB) |
|-----------|-----------|-----------------|-------|-----------|---------|
| 0 | 0 | 1 | 0.021 | 0 | |
| 0 | 1 | 1 | 0.022 | 0 | |
| 1 | 0 | 1 | 0.023 | 0 | 31 |
| 1 | 1 | 1 | 0.521 | 1 | |

### 3.3. OR

When the combination of input beams (01, 10, or 11) is inserted with REF at the same phase of 180°, the result of the $P_{out}$ becomes a '1'. Thus, the OR logic function between the two input beams is realized. Figure 7 depicts the OR field intensity distributions, using the proposed waveguide at 1.55 μm.

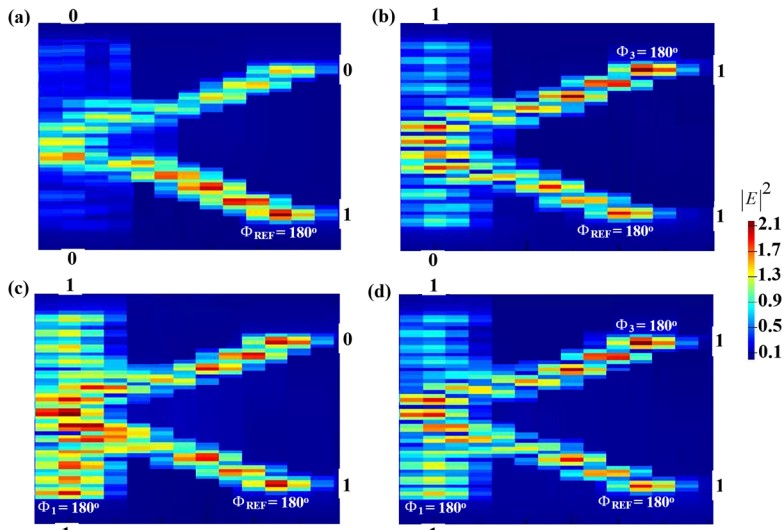

**Figure 7.** OR field intensity distributions, using K-shaped silicon-on-silica waveguide at 1.55 μm: (**a**) '00' input, (**b**) '01' input, (**c**) '10' input, and (**d**) '11' input.

The suggested waveguide obtains a high CR = 33.73 dB due to the significant difference between the mean peak powers of '1' and '0'. Table 4 provides an overview of the outcomes of the OR simulation at 1.55 μm, in terms of T and CR.

**Table 4.** OR simulation outcomes ($T_{th}$ = 0.12).

| $P_{in1}$ | $P_{in3}$ | $P_{in2}$ (REF) | T | $P_{out}$ | CR (dB) |
|---|---|---|---|---|---|
| 0 | 0 | 1 | 0.021 | 0 | |
| 0 | 1 | 1 | 0.464 | 1 | |
| 1 | 0 | 1 | 0.852 | 1 | 33.73 |
| 1 | 1 | 1 | 0.521 | 1 | |

The REF is essential for realizing the XOR, AND, and OR operations. Therefore, using the suggested waveguide at 1.55 μm, we have compared the performance of these three operations in terms of CR in the presence of the REF beam (i.e., REF = '1') and the absence of it, meaning there is no input beam injected into $P_{in2}$. Table 5 indicates the necessity of using the REF to obtain higher CRs.

**Table 5.** CR with and without REF.

| Operation | CR (dB) with REF | CR (dB) without REF |
|---|---|---|
| XOR | 34 | 7.1 |
| AND | 31 | 6.4 |
| OR | 33.73 | 7 |

*3.4. NOT*

To carry out all inverted logic operations, including NOT, NOR, NAND, and XNOR, a clock light (Clk) with an angle of 0° must be sent into the proposed waveguide from $P_{in1}$ of Figure 1. The Clk introduces an additional phase shift on the traveling beams, which changes the waveguide balance and results in an output. One beam is injected into $P_{in3}$ at an angle of 180° to perform the NOT operation. Due to the destructive interference that occurs as a result of the input beams' various phase conditions, when $P_{in3}$ is set to '1', $P_{out}$ produces a logical '0' (i.e., T < $T_{th}$). When $P_{in3}$ is 'OFF', the Clk (all '1's) outputs a logical '1' (i.e., T > $T_{th}$) at $P_{out}$, instead of going through a differencing phase. In this manner, the NOT gate is performed. Using a K-shaped silicon-on-silica waveguide, Figure 8 illustrates the NOT field intensity distributions at 1.55 μm.

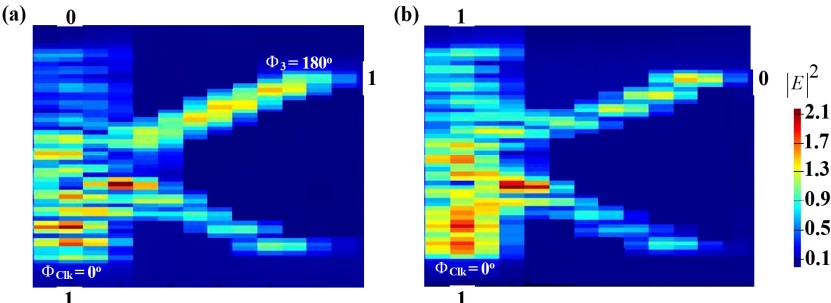

**Figure 8.** NOT field intensity distributions, using K-shaped silicon-on-silica waveguide at 1.55 μm: (**a**) '1' input and (**b**) '0' input.

The suggested waveguide results in a high CR = 30.5 dB for NOT operation. Table 6 provides a summary of the outcomes of the NOT simulation, using the proposed waveguide at 1.55 μm.

**Table 6.** NOT simulation outcomes ($T_{th}$ = 0.12).

| $P_{in1}$ (Clk) | $P_{in3}$ | T | $P_{out}$ | CR (dB) |
|---|---|---|---|---|
| 1 | 1 | 0.032 | 0 | |
| 1 | 0 | 0.675 | 1 | 30.5 |

### 3.5. NOR

Two beams are launched into $P_{in2}$ and $P_{in3}$ to perform the NOR (NOT-OR) operation, and $P_{in1}$ is launched with Clk (all '1's), as shown in Figure 1. When the input beams (01, 10, or 11) are combined and injected at different angles, destructive interference results in a logical '0' at $P_{out}$. If the launched beams' combination is (00), the Clk beam with $\Phi_{Clk} = 0°$ will cancel the phase balance of the three inputs, resulting in a logical '1' at $P_{out}$. Thus, the NOR logic operation is realized, as shown in Figure 9.

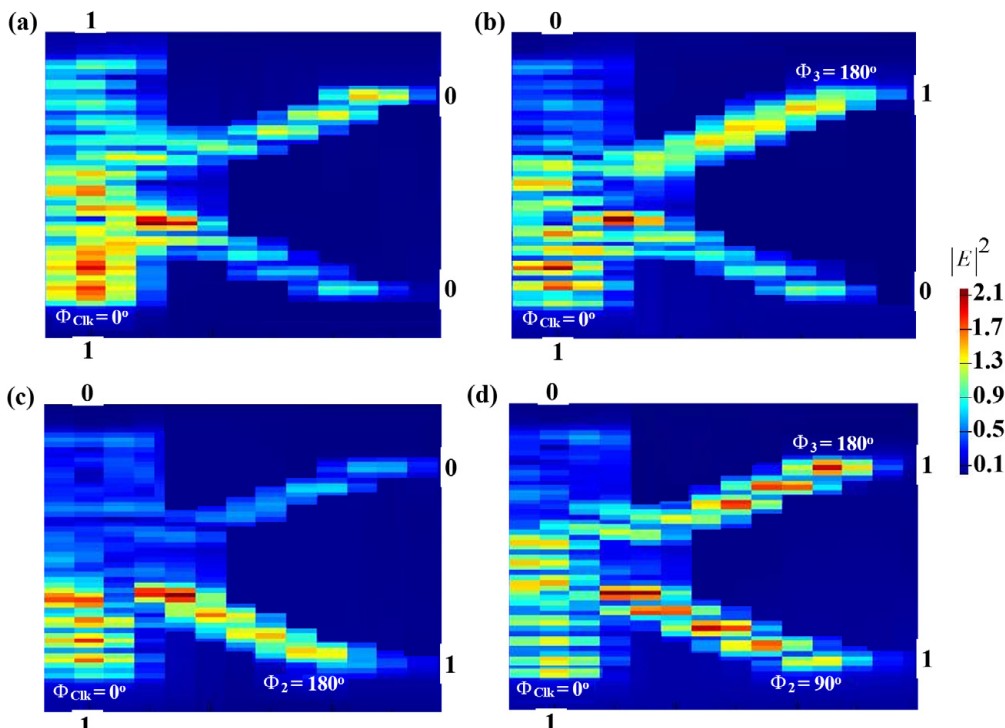

**Figure 9.** NOR field intensity distributions, using K-shaped silicon-on-silica waveguide at 1.55 μm: (**a**) '00' input, (**b**) '01' input, (**c**) '10' input, and (**d**) '11' input.

The suggested waveguide achieves a high CR = 33 dB for the NOR operation, as a result of the large disparity between $P_{mean}^1$ and $P_{mean}^0$. A summary of the simulation outcomes for this logic operation is given in Table 7.

**Table 7.** NOR simulation outcomes ($T_{th}$ = 0.12).

| $P_{in1}$ (Clk) | $P_{in2}$ | $P_{in3}$ | T | $P_{out}$ | CR (dB) |
|---|---|---|---|---|---|
| 1 | 0 | 0 | 0.675 | 1 | |
| 1 | 0 | 1 | 0.032 | 0 | |
| 1 | 1 | 0 | 0.022 | 0 | 33 |
| 1 | 1 | 1 | 0.022 | 0 | |

### 3.6. NAND

The NAND (NOT-AND) can be produced by injecting the Clk into $P_{in1}$ and the other two beams into $P_{in2}$ and $P_{in3}$, respectively. When both $P_{in2}$ and $P_{in3}$ are 'OFF' (i.e., 00), the

Clk with a $\Phi_{Clk} = 0°$ cancels the phase balance of the three inputs, causing $P_{out}$ to become '1'. Constructive interference simply occurs when Clk and (01, 10) are launched at the same angle of 0°, yielding an output of '1'. A '0' output is produced when (11) is launched with Clk at various phases, such as $\Phi_2 = 90°$, $\Phi_3 = 180°$, and $\Phi_{Clk} = 0°$, as illustrated in Figure 10.

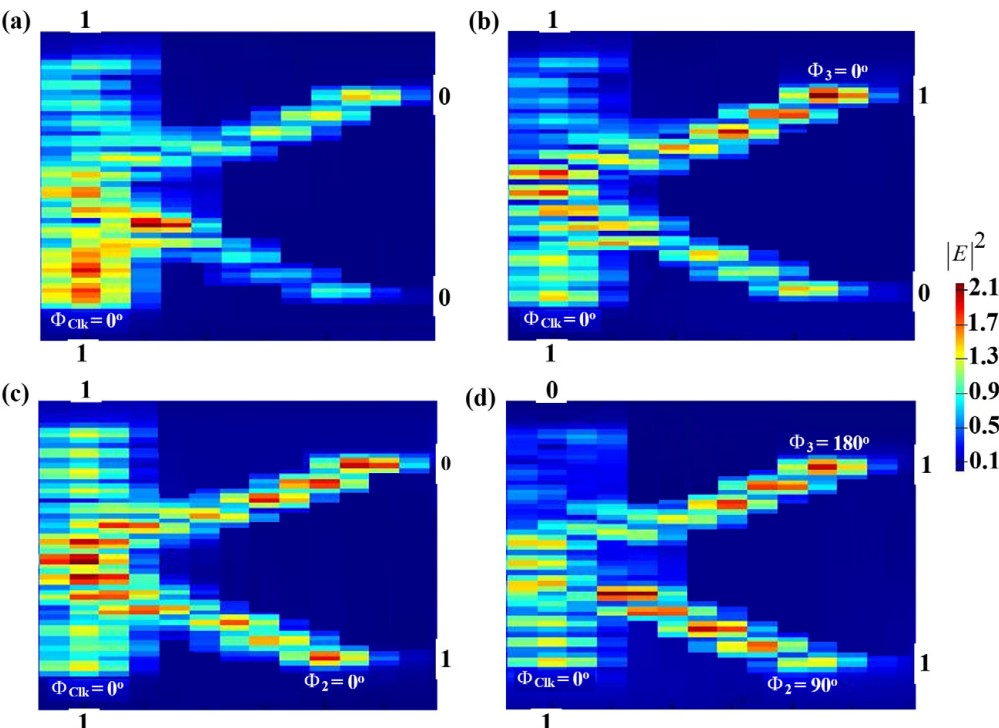

**Figure 10.** NAND field intensity distributions, using K-shaped silicon-on-silica waveguide at 1.55 μm: (**a**) '00' input, (**b**) '01' input, (**c**) '10' input, and (**d**) '11' input.

A summary of the NAND simulation outcomes utilizing the suggested waveguide, which achieves a high CR = 34 dB at 1.55 μm, is shown in Table 8.

**Table 8.** NAND simulation outcomes ($T_{th}$ = 0.12).

| $P_{in1}$ (Clk) | $P_{in2}$ | $P_{in3}$ | T | $P_{out}$ | CR (dB) |
|---|---|---|---|---|---|
| 1 | 0 | 0 | 0.675 | 1 | |
| 1 | 0 | 1 | 0.464 | 1 | |
| 1 | 1 | 0 | 0.852 | 1 | 34 |
| 1 | 1 | 1 | 0.022 | 0 | |

*3.7. XNOR*

Similar to the NOR and NAND operations, the Clk enters $P_{in1}$ to create the XNOR (exclusive-XOR) logic function, while the other two beams are injected from $P_{in2}$ and $P_{in3}$, respectively. Constructive interference causes $P_{out}$ to emit a '1' when the combination of the input beams (11) is introduced with the Clk at the same phase of 0°. In contrast, $P_{out}$ produces a '0' when the input beams' combinations, (01) or (10), are inserted with a different phase, as depicted in Figure 11.

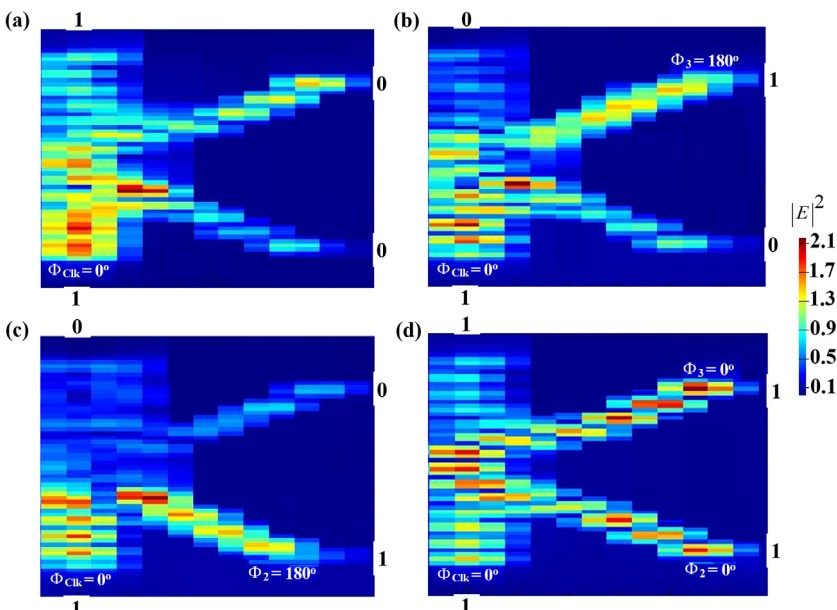

**Figure 11.** XNOR field intensity distributions, using K-shaped silicon-on-silica waveguide at 1.55 μm: (**a**) '00' input, (**b**) '01' input, (**c**) '10' input, and (**d**) '11' input.

Table 9 summarizes the XNOR simulation outcomes with a high CR = 31 dB, using the suggested waveguide.

**Table 9.** XNOR simulation outcomes ($T_{th}$ = 0.12).

| $P_{in1}$ (Clk) | $P_{in2}$ | $P_{in3}$ | T | $P_{out}$ | CR (dB) |
|:---:|:---:|:---:|:---:|:---:|:---:|
| 1 | 0 | 0 | 0.675 | 1 | |
| 1 | 0 | 1 | 0.032 | 0 | |
| 1 | 1 | 0 | 0.022 | 0 | 31 |
| 1 | 1 | 1 | 0.521 | 1 | |

The Nyquist formula gives the speed of a transmission system as $2B \log_2[M]$ [16], where M is the total number of signal levels, and B is the optical bandwidth, which is defined as $B = (c/\lambda^2)\Delta\lambda$, where c is the speed of light in vacuum, λ = 1.55 μm is the optical carrier wavelength, and Δλ is the signal's spectral width. Note $\log_2[M]$ is in a binary form, i.e., $\log_2[M] = \log[M]/\log[2]$. This means that in our case, where B = 30 GHz and for four signal levels (00, 01, 10, and 11), the predicted speed is 120 Gb/s.

Silicon and silica components are readily available, making it easier and more affordable to build the suggested waveguide. As a result, assuming that the necessary technology is available and that the major outcomes of this simulation are valid, the experimental verification of the suggested waveguide may be completed. This is a technology problem that can be resolved in practice, so it is not a crucial obstacle. Several logic operations, on the other hand, have been experimentally implemented based on various optical waveguides and components [31–38].

Table 10 compares the functionality of the considered waveguide, for realizing the intended logic operations at various wavelengths, to that of several waveguides reported on the same topic. This table suggests that the proposed waveguide can achieve faster logic operations with higher CRs than the other listed schemes.

**Table 10.** At various wavelengths, a comparison of our design and other waveguide-based logic function designs.

| Operations | Design | Wavelength (nm) | CR (dB) | Ref. |
|---|---|---|---|---|
| XOR, AND, OR, NOR, NAND, XNOR | Dielectric-loaded waveguides | 471 | 24.41–33.39 | [16] |
| OR, NOT, AND, XOR | Metallic waveguide arrays | 632.8 | 9.3–20 | [17] |
| NOT, XOR, AND, OR, NOR, NAND, XNOR | Nanoring insulator–metal–insulator waveguides | 1550 | −1.1–18.75 | [18] |
| NOT, XOR, AND, OR, NOR, NAND, XNOR | Dielectric–metal–dielectric design | 900 and 1330 | 5.37–22 | [19] |
| AND, OR, NAND, NOR, XOR, Fan-Out, Half adder, Full adder | Photonic crystal circiuts | 1550 | 5.54–11.56 | [20] |
| XOR, AND, OR, NOT, NOR, XNOR, NAND | K-shaped silicon waveguides | 1550 | 30.5–34 | This work |

## 4. Conclusions

Using K-shaped silicon-on-silica waveguides, seven fundamental logic operations, including XOR, AND, OR, NOT, NOR, NAND, and XNOR, were simulated at the 1.55 µm telecommunications wavelength. These operations were simulated by means of FDTD solutions obtained in commercially available software. The correct execution of these logic operations relies on the constructive and destructive interferences that are caused by the suitable phase difference of the launched optical input beams. Compared to other waveguides reported for the same purpose, the suggested K-shaped waveguide achieves logic operations with a higher contrast ratio and operating speed.

**Author Contributions:** Conceptualization, A.K.; data curation, A.K.; formal analysis, A.K.; funding acquisition, A.K.; investigation, A.K.; methodology, A.K.; project administration, A.K.; resources, A.K.; software, A.K.; supervision, K.E.Z.; writing—original draft, A.K.; writing—review and editing, K.E.Z. All authors have read and agreed to the published version of the manuscript.

**Funding:** This research received no external funding.

**Data Availability Statement:** Not applicable.

**Acknowledgments:** A.K. thanks the Chinese Academy of Sciences President's International Fellowship Initiative (Grant No. 2022VMB0013) for the support of this work.

**Conflicts of Interest:** The authors declare no conflict of interest.

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
