# Peer review of "K-Shaped Silicon Waveguides for Logic Operations at 1.55 μm"

_electronics, doi:10.3390/electronics11223748_

Round 1

Reviewer 1 Report

Authors presented K-shaped for logic operations including XOR, AND, OR, NOT, XNOR and NAND. The paper needs extensive revision.

All the field intensity figures presented are blurred and need to be replaced with good-quality figures.

The meshing used needs to be on a higher side to remove the blurred part in the K-shape waveguide Electric field intensity figures.

The comparison presented is with old references and there should be new references added to show the advancement.

The writing of the paper needs to be improved.

the authors are encouraged to resubmit the manuscript after making these extensive revisions.

Author Response

Authors’ response

Manuscript ID: Electronics-2018622

Title: K-shaped silicon waveguides for logic operations at 1.55 μm

Electronics

Thank you for giving us the opportunity to revise and resubmit the above manuscript.

The manuscript has been revised according to the reviewers’ suggestions, as described in the attached response (noted in red color for Reviewer #1, blue color for Reviewer #2, and green color for Reviewer #3 are the changes that have been incorporated in the manuscript).

We hope that we have met all the reviewers’ suggestions and that you will, therefore, find our manuscript appropriate for publication in Electronics.

Sincerely,

Amer Kotb

Kyriakos Zoiros

Reviewer 2 Report

Silicon waveguides have wide applications, particularly for all-optical networks. Increasingly sophisticated structures of photonic components based on waveguides have been developed, such as pigtail micro-rings, MZIs, Y splitters. Overall, this study developed an innovative design to simulate seven basic logic operations using the FDTD and K-shaped silicon waveguides operating at a communications wavelength of 1.55 μm and exhibited applicable performances. I have some comments or questions as followed, before considering this work for publishing.

1. For Figure 6 and Figure 9, some of the pictures, like Pin 3 of the second image in Figure 6, are not labeled with "0" since the port lacks input. All the figures should be in the uniform format. Additionally, it might be better to annotate all of the figures with (a) (b) (c), as this will help the readers to understand the figures better.

2. Page 8 Line 260, Several logic operations have been experimentally implemented based on various waveguides, “various waveguides” is not only in structure, but also in materials. Maybe here more articles could be cited to show the work here is special or even superior (e.g. Lab on a chip 20, 3815-3823, 2020. IEEE Photonics Technology Letters, 2019. 31(17): p. 1425-1428. Electronics letters, 2018. 54(4): p. 229-231)

3. The introduction part seems to be not sufficient to make clear the novelty or special contribution of this work compared with others’ work. It is only about the silicon photonic devices and signal processing tasks using logic gates, it would be more attractive if the article about the waveguides which are used in all-optical network can be cited and compared. Besides, page 2 Line 50, “the data better than other reported designs”, is it necessary to have some references to prove that your work performs better?(e.g., Ultra-compact, low RF power, 10 Gb/s silicon Mach-Zehnder modulator. Optics express, 2007. 15(25): p. 17106-17113.)

4. For table 1 and table 10, the operating wavelength for simulation is 1.0 μm, but the title show that it is at communication wavelength 1.55μm, is it contradictory?

5. Page 8 Line 253, B is the optical bandwidth, but the value taken in your calculation is 1.55 μm, which is wavelength rather than bandwidth. Is wavelength equal to the bandwidth? So is it really possible to actually reach a speed of 200 Gb/s?

6. In your simulation diagram, there are some input ports that are “0”, while showing strong field strength, it is necessary to explain the situation briefly. Maybe a finer grid provides a better view of the energy transfer at the coupling point of 3 input waveguides. It may be helpful to explain the situation mentioned above.

7. For Table 5, “without REF”, could you give more details? Is there no input, or is the phase 0?

8. Page 2 Line 67, “phase-matching” in article’s simulation is mentioned many times, maybe some theory can be provided to support your simulation and your conclusions of this paper.

Author Response

(The authors gave the same response as above.)

Author Response

(The authors gave the same response as above.)

Round 2

Reviewer 1 Report

The authors presented a "K-shaped silicon waveguides for logic operations at 1.55 μm" with a demonstration of all basic optical logic operations, including XOR, 12 AND, OR, NOT, NOR, XNOR, and NAND. The waveguide comprises of three waveguide strips, all made of silicon printed on silica. By adjusting 15 the phase of the incident beams, the pursued logic operations can be realized. To evaluate how well 16 the considered operations are performed, the contrast ratio (CR) is employed as figure-of-merit. 17 Compared to other reported waveguides, the suggested K-shaped waveguide achieves higher CRs 18 and a speed of the order of 120 Gb/s. The proposed manuscript is very interesting and can be accepted after the authors resolve the following concerns:

1.) The abstract conveys very little information related to the study. Further detailed information related to how the study is conducted with materials and other important details should be included in the abstract.

2.) There are so many novel ways to realize logic gates such as:

Implementation of novel boolean logic gates for IMPLICATION and XOR functions using riboregulators. Bioengineered13(1), 1235-1248

Design of Cost-Efficient Graphene Metasurface based Pregnancy Test with NOR Gate Realization and Parametric Optimization," in IEEE Sensors Journal, 2022, doi: 10.1109/JSEN.2022.3218797.

Two-qubit logic gates based on the ultrafast spin transfer in π-conjugated graphene nanoflakes. Carbon193, 195-204.

3.) All figures should be self-explanatory.

4.) I think Fig. 5-11 are wrong as it doesn't agree with the corresponding table. Please elaborate on this. And provide the correct 0 1 labeling with p1 to P3 labeling.

5.) How the proposed structure is comparatively novel and its advantage over other structure should be mentioned.

Author Response

N/A
